# Comparing mutational pathways to lopinavir resistance in HIV-1 subtypes B versus C

**Susana Posada-Céspedes**[1,2], **Gert Van Zyl**[3,4], **Hesam Montazeri**[5], **Jack Kuipers**[1,2], **Soo-Yon Rhee**[6], **Roger Kouyos**[7,8], **Huldrych F. Günthard**[7,8], **Niko Beerenwinkel**[1,2]*

**1** Department of Biosystems Science and Engineering, ETH Zurich, Basel, Switzerland, **2** SIB Swiss Institute of Bioinformatics, Basel, Switzerland, **3** Division of Medical Virology, Faculty of Medicine and Health Sciences, Stellenbosch University, Cape Town, South Africa, **4** National Health Laboratory Service, Cape Town, South Africa, **5** Department of Bioinformatics, Institute of Biochemistry and Biophysics, University of Tehran, Tehran, Iran, **6** Department of Medicine, Stanford University, Stanford, California, United States of America, **7** Division of Infectious Diseases and Hospital Epidemiology, University Hospital Zurich, Zurich, Switzerland, **8** Institute of Medical Virology, University of Zurich, Zurich, Switzerland

* niko.beerenwinkel@bsse.ethz.ch

**Data Availability Statement:** Patient identifiers and isolate year of sequences retrieved form the Stanford HIV Drug Resistance Database are provided in the Supporting information files. A

## Abstract

Although combination antiretroviral therapies seem to be effective at controlling HIV-1 infections regardless of the viral subtype, there is increasing evidence for subtype-specific drug resistance mutations. The order and rates at which resistance mutations accumulate in different subtypes also remain poorly understood. Most of this knowledge is derived from studies of subtype B genotypes, despite not being the most abundant subtype worldwide. Here, we present a methodology for the comparison of mutational networks in different HIV-1 subtypes, based on Hidden Conjunctive Bayesian Networks (H-CBN), a probabilistic model for inferring mutational networks from cross-sectional genotype data. We introduce a Monte Carlo sampling scheme for learning H-CBN models for a larger number of resistance mutations and develop a statistical test to assess differences in the inferred mutational networks between two groups. We apply this method to infer the temporal progression of mutations conferring resistance to the protease inhibitor lopinavir in a large cross-sectional cohort of HIV-1 subtype C genotypes from South Africa, as well as to a data set of subtype B genotypes obtained from the Stanford HIV Drug Resistance Database and the Swiss HIV Cohort Study. We find strong support for different initial mutational events in the protease, namely at residue 46 in subtype B and at residue 82 in subtype C. The inferred mutational networks for subtype B versus C are significantly different sharing only five constraints on the order of accumulating mutations with mutation at residue 54 as the parental event. The results also suggest that mutations can accumulate along various alternative paths within subtypes, as opposed to a unique total temporal ordering. Beyond HIV drug resistance, the statistical methodology is applicable more generally for the comparison of inferred mutational networks between any two groups.

deposition of the HIV sequence data obtained from the Swiss HIV Cohort Study (SHCS) in an open database is not possible at this time. This is due to the representativeness of the dataset, the sensitivities associated with HIV infections, and to protect the privacy of patients enrolled in the study. The code needed to conduct the data analysis is available at https://github.com/cbg-ethz/MC-CBN and https://github.com/cbg-ethz/H-CBN2-comparison-test.

**Funding:** The SHCS and its drug resistance database was supported by the Swiss National Science Foundation (grant number 179 571 to H. F. G.) and was financed within the framework of the SHCS, supported by the Swiss National Science Foundation (grant number 148522), by the SHCS Research Foundation and by the Yvonne Jacob Foundation (to H. F. G.). S.R. was supported in part by the National Institute of Allergy and Infectious Diseases (NIAID) of the National Institute of Health (NIH) (award number AI136618). The funders had no role in study design, data collection and analysis, decision to publish, or preparation of the manuscript.

**Competing interests:** I have read the journal's policy and the authors of this manuscript have the following competing interests: H. F. G. has received unrestricted research grants from Gilead Sciences and Roche, fees for data and safety monitoring board membership from Merck; consulting/advisory board membership fees from Gilead Sciences, Viiv and Merck; and grants from SystemsX, and the National Institutes of Health.

## Author summary

There is a disparity in the distribution of infections by HIV-1 subtype in the world. Subtype B is predominant in America, Australia and western and central Europe, and most therapeutic strategies are based on research and clinical studies on this subtype. However, non-B subtypes represent the majority of global HIV-1 infections; e.g., subtype C alone accounts for nearly half of all HIV-1 infections. We present a statistical framework enabling the comparison of patterns of accumulating mutations in different HIV-1 subtypes. Specifically, we compare the temporal ordering of lopinavir resistance mutations in HIV-1 subtypes B versus C. To this end, we combine the Hidden Conjunctive Bayesian Network (H-CBN) model with an approximate inference scheme enabling comparisons of larger networks. We show that the development of resistance to lopinavir differs significantly between subtypes B and C, such that findings based on subtype B sequences can not always be applied to sybtype C. The described methodology is suitable for comparing different subgroups in the context of other evolutionary processes.

## Introduction

HIV-1 infections are clinically manageable by combining two or more antiretroviral drugs [1], but the accumulation of drug resistance mutations, a process driven by the evolutionary escape dynamics of HIV-1, still limits their success [2, 3]. These therapies, although largely developed based on studies of HIV-1 subtype B viruses, have been shown to be effective in controlling infection without subtype-specific differences [4]. However, there is increasing evidence of differences in mutation patterns and evolutionary rates among subtypes [4–10], but knowledge on subtype-specific mutational pathways is very limited.

Here, we investigate the rate and partial order of accumulation of drug resistance mutations in different HIV-1 subtypes. Specifically, we compare mutational networks to lopinavir resistance in HIV-1 subtypes B versus C. Although HIV-1 subtype B is the best studied and most prevalent subtype in America, Australia and western and central Europe, subtype C alone accounts for nearly half of all HIV infections worldwide [11, 12]. It is therefore important to understand whether the evolution of drug resistance in subtype C proceeds in a similar fashion as for subtype B.

The accumulation of advantageous mutations under the selective pressure of several antiretroviral drugs have been studied by sequencing the HIV-1 genome derived from patients over the course of treatment [13–20]. However, such longitudinal data are not available for most antiretroviral therapies. To leverage information from large cohorts and cross-sectional studies, different statistical models have been proposed to investigate inter-dependencies of mutations. On the one hand, approaches such as Bayesian networks [21–23] and Cox proportional-hazards models [24] provide insights into statistical dependencies between resistance mutations without explicitly modeling the ordering among such mutations. On the other hand, order-aware approaches for investigating evolutionary pathways leading to drug resistance include mutagenetic trees [25, 26]; discrete and continuous-time Conjunctive Bayesian Networks (CBN) [27, 28]; and Suppes-Bayes Causal Networks (e.g., CAPRI [29, 30]). Most of the aforementioned methods have been applied to study drug resistance mutations in HIV-1 subtype B infections. As an exception, Deforche *et al.* [21, 22] combined observations from various subtypes to investigate dependencies among resistance mutation and polymorphisms using Bayesian networks. The inferred network was used to explain the lower prevalence of

protease mutation 30N in subtypes G and A as compared to subtype B through an interaction with the polymorphic locus 89L/M.

To infer subtype-specific lopinavir mutational networks, here we use the Hidden Conjunctive Bayesian Network (H-CBN) [31], an extension of the continuous-time Conjunctive Bayesian Network (CT-CBN) accounting for observational errors. The CT-CBN model encodes constraints on the temporal ordering among mutations by assuming that the occurrence of genetic events can depend on the occurrence of predecessors. While in tree-based models the number of direct predecessors is constrained to be at most one, this assumption is relaxed in the CT-CBN model, where multiple predecessors are allowed. Therefore, CT-CBNs can be viewed as a generalization of mutagenetic trees and in practice also outperform them [27, 32]. In other comparative studies using data simulated under various fitness landscapes, H-CBN models typically provide comparable or better performance to competitors such as CAPRI [33] and trees [33, 34] in reconstructing the mutational networks for most of the evaluated metrics. In addition, the H-CBN model incorporates a superior error model as described below.

The partial order among mutations is inferred from observed viral genotypes. However, genotyping is error-prone and, in fact, the H-CBN model was developed as an extension of the CT-CBN to account for sequencing errors [31] and to improve the simple mixture error model of the original CT-CBN [28] which featured two model components to distinguish signal from noise. The CT-CBN model has been applied to learn mutational networks in HIV under different selective pressures. The data sets originally analyzed included at most nine resistance mutations, but Montazari *et al.* [35] presented the MC-CBN Monte Carlo expectation-maximization algorithm for parameter estimation of the mixture model applicable to hundreds of mutations. Yet, the mixture error model has several limitations. Every genotype that violates the ordering constraints is assumed to occur with equal probability regardless of, e.g., the number of violations. Moreover, as the number of mutations increases the chance of obtaining an error-free genotype decreases rapidly. For instance, with a 1% per locus error rate and 64 mutations, we expect only around 53% of the genotypes to be correct. The mutation network is, however, inferred exclusively from the portion of the data assigned to the signal component of the mixture model, which can quickly result in a large portion of the data being discarded.

In their H-CBN extension of the CT-CBN, Gerstung *et al.* [31] introduced latent variables to explicitly model the noisy observation process, which is parameterized by a per-locus error rate. In contrast to the mixture model, genetic events that apparently violate the ordering constraints can be explained by the latent variables, and the assumption that all violations are equally likely is relaxed. Moreover, instead of using only compatible genotypes to infer the maximum likelihood network as in the mixture model, the H-CBN uses all observed genotypes in a weighted fashion. Inference of the H-CBN model has been implemented via maximum likelihood estimation, but the time complexity of the likelihood computation is exponential in the number of mutations. In practice, computation quickly becomes impractical as the number of genetic events grows beyond 14 mutations, whereas our study involves up to 20 mutations.

Here, we take advantage of the improved error model of the H-CBN, but address its limitation regarding scalability in the number of mutations by employing an approximation scheme for the estimation of model parameters. We assess the performance of our method on simulated data and compare it to the original H-CBN method. Furthermore, we incorporate an adaptive simulated annealing algorithm to infer the maximum likelihood mutational network from the data, including different moves to explore the discrete space of networks. We compare the accuracy of the networks reconstructed using this method to the MC-CBN method which implements a mixture error model. The resulting model and inference methods, called H-CBN2, are implemented as part of the MC-CBN R-package available at https://github.com/cbg-ethz/MC-CBN.

We use the H-CBN2 method to infer evolutionary pathways to lopinavir resistance in HIV-1 subtypes B and C, confirming previous knowledge on frequently observed patterns of resistance-conferring mutations in response to lopinavir treatment [36, 37]. We also devise a statistical test to assess the similarity between two CBN models, which is available at https://github.com/cbg-ethz/H-CBN2-comparison-test. When applied to subtypes B versus C, we find significant differences in their mutational networks.

## Materials and methods

### Ethics statement

The Swiss HIV Cohort Study (SHCS) has been approved by the ethical committees of the participating institutions: Kantonale Ethikkommission Bern; Ethikkommission beider Basel; Comite departemental d'éthique des specialites medicales et de medicine communataire et de premier recours, Hôpitaux Cantonale de Genève; Commission cantonale d'éthique de la recherche sur l'être humain, Canton de Vaud, Lausanne; Repubblica e Cantone Ticino–Comitato Ethico Cantonale; Ethikkommission des Kantons St. Gallen; and Kantonale Ethikkommission Zürich. Written informed consent has been obtained from all participants.

### Hidden conjunctive Bayesian network

Methods described in this section are organized as follows. We first recapitulate the probabilistic graphical model underlying this work, the H-CBN. Second, we introduce a new parameter inference method for the H-CBN model, as well as an improved structure learning algorithm based on adaptive simulated annealing. Third, we develop a statistical test to assess structural differences between two CBN models.

CBNs are probabilistic graphical models, in which a directed acyclic graph (DAG) represents the ordering constraints among genetic events [27]. In the CT-CBN, the time between genetic events is modeled by independent exponential distributions [28]. The H-CBN extends the CT-CBN model by introducing hidden variables to model the error-prone observational process [31].

Formally, the CT-CBN is defined by a partially ordered set (poset) of genetic events, or mutations, and a rate for each mutation to occur. A poset $(P, \prec)$ consists of a set $P$ of size $p = |P|$ and a binary relation $\prec$. The relation $l \prec k$ indicates that mutation $l$ must take place before $k$. Further, a relation $l \prec k$ is a cover relation if $l \prec z \prec k$ implies $z = l$ or $z = k$. Drawing a directed edge from node $l$ to node $k$ for every cover relation $l \prec k$ yields a DAG which is transitively reduced and uniquely represents the poset [28]. It is therefore sufficient to consider transitively reduced DAGs only.

A genotype is a subset of genetic events of $P$, represented by a binary vector $x = (x_1, \ldots, x_p)$, where $x_j = 1$ indicates that mutation $j$ has occurred. A genotype $x$ is called compatible with the poset $P$ if $(x_l, x_k) \neq (0, 1)$ for all cover relations $l \prec k$. The collection of all genotypes compatible with $P$ is the genotype lattice $J(P)$, which defines the space of all feasible mutational patterns.

The waiting time to each mutation $j$ is represented by a random variable $T_j$. Their joint distribution is defined recursively as

$$T_j \sim Z_j + \max_{u \in \mathrm{pa}(j)} T_u, \quad Z_j \sim \mathsf{Exp}(\lambda_j), \tag{1}$$

where $\mathrm{pa}(j)$ denotes the set of parents of $j$ in the DAG, i.e., the set of mutations which precede mutation $j$. The random variable $Z_j$ is exponentially distributed with rate $\lambda_j$ and accounts for the time elapsed for generating and fixating mutation $j$, after its predecessors have occurred.

The time at which every individual mutation emerges is generally unknown. Instead, patients are monitored with certain regularity, and oftentimes when the viral load increases, the virus population is sequenced. The sampling time is generally also unknown and typically differs among patients. To account for this uncertainty, an exponentially distributed random variable $T_s \sim \text{Exp}(\lambda_s)$ is introduced. Hence, the observed data is censored, and a mutation $j$ occurred if and only if its waiting time $t_j$ was smaller than the sampling time $t_s$, i.e., $x_j = 1$ if $t_j < t_s$ and $x_j = 0$ otherwise. The model is not identifiable as long as the rate $\lambda_s$ is unknown. Therefore, unless known, this scaling factor is set to $\lambda_s = 1$ [28, 31].

There is another hidden process, namely the generation of viral genotype data. In the H-CBN model, a variable $Y$ is introduced to denote the observed genotype, an error-bearing version of the true genotype $X$ [31]. Assuming errors are independent and identically distributed across mutations, the probability of observing genotype $Y$ given the true underlying genotype $X$ is defined by a Bernoulli process with parameter $\epsilon$, the per-locus error probability.

## Parameter estimation via Monte Carlo Expectation Maximization

Owing to censoring of mutation times and unobserved true genotypes, the Expectation Maximization algorithm (EM) has been previously used to obtain maximum likelihood estimates of model parameters $\epsilon$ and $\lambda_j$, $j = 1, \ldots, p$ [31]. To address the limitation on the scalability in the number of mutations, we develop a Monte Carlo Expectation Maximization algorithm (MCEM) to jointly estimate the error rate ($\epsilon$) and the conditional evolutionary rate parameters ($\lambda_j$, $j = 1, \ldots, p$) for a given poset $P$.

In the expectation step (E step) of the MCEM algorithm, we estimate the expected value of the complete-data log-likelihood $\ell_{\mathcal{X},\mathcal{Z},\mathcal{Y}}(\lambda, \epsilon)$ with respect to the current conditional distribution of the hidden data (i.e., the unobserved true genotypes $\mathcal{X} = (X^{(1)}, \ldots, X^{(N)})$ and mutation times $\mathcal{Z} = (Z^{(1)}, \ldots, Z^{(N)})$), given the observed genotypes $\mathcal{Y}$, as well as the current estimates of the parameters $\lambda^{(k)}$ and $\epsilon^{(k)}$

$$
\mathbb{E}_{\mathcal{X},\mathcal{Z}|(\mathcal{Y},\lambda^{(k)},\epsilon^{(k)})}[\ell_{\mathcal{Y},\mathcal{X},\mathcal{Z}}(\lambda, \epsilon)] =
$$
$$
\sum_{i=1}^{N} \sum_{x^{(i)} \in J(P)} \int_{\mathbb{R}_{\geq 0}^{p+1}} \ell_{\mathcal{Y},\mathcal{X},\mathcal{Z}}(\lambda, \epsilon) f_{X,Z}\left(x^{(i)}, z^{(i)} \,\Big|\, Y = y^{(i)}; \lambda^{(k)}, \epsilon^{(k)}\right) dz^{(i)}, \tag{2}
$$

where $k$ denotes the current MCEM iteration (see S1 Text for details).

For small H-CBN models, this integral has been computed by decomposing it into a sum of integrals over all possible maximal chains in the genotype lattice [28, 31]. However, the number of maximal chains is $p!$ in the worst case, where $p$ is the number of mutations. Moreover, the summation over all possible genotypes in $J(P)$ is bounded by the total number of unobserved true (binary) genotypes: $2^p$. For moderate to large numbers of mutations, the exact computation of the expected value thus becomes computationally infeasible. To overcome this limitation, we approximate the expected value, Eq (2), using importance sampling. The general idea is to generate $L$ samples of the unobserved true genotypes $x$ and the mutation times $z$ from a proposal distribution $Q(x, z)$. Then,

$$
\mathbb{E}_{\mathcal{X},\mathcal{Z}|(\mathcal{Y},\lambda^{(k)},\epsilon^{(k)})}[\ell_{\mathcal{Y},\mathcal{X},\mathcal{Z}}(\lambda, \epsilon)] \approx
$$
$$
\frac{1}{L} \sum_{i=1}^{N} \sum_{l=1}^{L} \frac{1}{Q(x_l^{(i)}, z_l^{(i)})} \ell_{\mathcal{Y},\mathcal{X},\mathcal{Z}}(\lambda, \epsilon) f_{X,Z}\left(x^{(i)}, z^{(i)} \,\Big|\, Y = y^{(i)}; \lambda^{(k)}, \epsilon^{(k)}\right). \tag{3}
$$

Intuitively, we would like to draw samples from the important region, e.g., samples that are likely to have given rise to the observed data. We use two types of importance sampling

schemes, which we refer to as the forward and backward sampling, and implement and compare several variations of them (see next subsections).

In the maximization step (M step), we are concerned with maximizing Eq (2) with respect to the parameters $\epsilon$ and $\lambda_j$, $j = 1, \ldots, p$. The maximum likelihood (ML) estimate $\hat{\epsilon}$ of the error rate $\epsilon$ is found to be the conditional expectation of the sufficient statistic $d_H(X, Y)$ obtained in the E-step,

$$\hat{\epsilon}^{(k)} = \frac{1}{N} \sum_{i=1}^{N} \mathbb{E}_{X,Z|Y,\lambda^{(k)},\epsilon^{(k)}} \left[ \frac{1}{p} d_H(x^{(i)}, y^{(i)}) \right]. \tag{4}$$

Similarly, the ML estimate for the rate parameters $\hat{\lambda}_j$ are,

$$[\hat{\lambda}_j^{(k)}]^{-1} = \frac{1}{N} \sum_{i=1}^{N} \mathbb{E}_{X,Z|Y,\lambda^{(k)},\epsilon^{(k)}} \left[ z_j^{(i)} \right]. \tag{5}$$

**Forward sampling.**   Assume the rate parameters $\lambda$ and the poset $P$ are known. We generate a candidate error-free genotype $x$ by sampling the mutation and sampling times $z = (z_1, \ldots, z_p, t_s)$ from the corresponding exponential distributions as follows

$$z_j \sim \mathsf{Exp}(\lambda_j), \; j = 1, \ldots, p, \quad t_s \sim \mathsf{Exp}(\lambda_s).$$

To determine the waiting times $t = (t_1, \ldots, t_p)$ we set $t_j = z_j + \max_{u \in \mathrm{pa}(j)} t_u$. Whenever, the waiting time $t_j$ for mutation $j$ is smaller than the sampling time $t_s$, we record that the mutation $j$ has been observed. If we do this for every mutation $j$, we obtain a sample of an error-free genotype $x = (x_1, \ldots, x_p)$. We draw samples by traversing the DAG in topological order to ensure that we compute $t_u$ for all $u \in \mathrm{pa}(j)$ before visiting any dependent mutation $j$. Because we do not know the rate parameters $\lambda$, nor the poset $P$, in each iteration of the MCEM algorithm, we use their current estimates, $\lambda^{(k)}$ and $P^{(k)}$.

For each observed genotype $y^{(i)}$, $i = 1, \ldots, N$, we draw $L$ samples using the forward sampling scheme described above. A sample is a tuple of waiting times and the corresponding error-free genotype. Because of the graph traversal and the loop over parents, the worst-case time-complexity of the forward sampling is $O(NLp^2)$. We note that the candidate hidden genotypes are generated without accounting for the observed data. Alternatively, we implement a second forward sampling scheme called forward-pool. In this case, for each iteration of the MCEM algorithm, we draw an initial pool of $K$ waiting times vectors $(t_j^{(l)}, j = 1, \ldots, p)$, with $K \gg L$, and for each observed genotype, we choose a subset of $L$ samples according to their similarity to the observed genotype as explained below. For each of the waiting times samples, we first construct the error-free genotype $x^{(l)}$ and then draw $L$ genotypes, each with probability

$$q_l = \frac{\epsilon^{d_H(y^{(i)}, x^{(l)})} (1 - \epsilon)^{p - d_H(y^{(i)}, x^{(l)})}}{\sum_{l=1}^{K} \epsilon^{d_H(y^{(i)}, x^{(l)})} (1 - \epsilon)^{p - d_H(y^{(i)}, x^{(l)})}}. \tag{6}$$

**Backward sampling.**   For the backward sampling, we construct the sample of candidate error-free genotypes $x^{(l)}$, $l = 1, \ldots, L$, based on the observed genotype $y^{(i)}$ and then sample the mutation times as

$$z_j \sim \begin{cases} \mathsf{TExp}(\lambda_j, \, 0, \, t_s - \max_{u \in \mathrm{pa}(j)} t_u) & \text{if } x_j = 1 \\ \\ \mathsf{Exp}(\lambda_j) & \text{otherwise,} \end{cases} \tag{7}$$

where TExp is a truncated exponential distribution. Montazeri *et al.* [35] have used Eq (7) to generate mutation times only from the compatible genotypes while using a mixture error model. Here, we extend this approach to also include sampling of the hidden layer modeling the genotyping errors, which enables us to account for all the observations.

We implement three variations of backward sampling to construct the sample of candidate hidden (true) genotypes. For the first strategy, we generate the genotypes $x^{(l)}$ by enumerating all compatible genotypes within Hamming distance $k$ of the observed genotype $y^{(i)}$, typically with $k \leq 3$. We then draw $L$ waiting-time vectors for each candidate genotype according to Eq (7). This sampling scheme is referred to as Hamming $k$-neighborhood sampling. In the second strategy, we sample candidate genotypes by altering individual mutations of the observed genotype using $p$ independent Bernoulli trials, one for each mutation $j = 1, \ldots, p$, with success probability equal to the current estimate of the error rate $\hat{\epsilon}^{(k)}$. We draw $L$ candidate genotypes some of which may be incompatible with the current poset $P^{(k)}$ and, thus, obtain a zero sampling weight; i.e., they do not contribute to the estimation of the model parameters. This sampling scheme is referred to as Bernoulli sampling. The third approach is a two-step scheme. First, we decide uniformly at random whether to (i) leave the genotype $y^{(i)}$ unperturbed, (ii) add, or (iii) remove a mutation. For (ii) and (iii), we draw a mutation from the set of mutations that can be added or removed, respectively. If we remove an event $j$, it is chosen with probability proportional to $\kappa_j = \frac{1}{\lambda_j} + \max_{l \in \text{pa}(j)} \kappa_l$, which corresponds to a greedy approximation of the time to mutation assuming that the process is dominated by the slowest predecessor in each reverse breadth-first search generation. The rationale is to remove mutations from the genotype $y^{(i)}$ that are likely to occur at later times with higher probability. On the other hand, if we add an event, it is chosen with a probability which is inversely proportional to the probability of being removed. In this case, we add mutations that can arise faster with higher probability. In the second step of this scheme, we ensure the genotype is compatible with the current poset $P^{(k)}$ by adding or removing all incompatible mutations. This sampling scheme is referred to as the backward-add/remove (backward-AR) sampling.

**Evaluation of sampling schemes.**   We evaluate the accuracy of the different approximation schemes by computing the probability of a genotype $y^{(i)}$ and comparing it to the exact solution [31]. Since $\Pr(Y = y^{(i)})$ are the factors of the likelihood, we are assessing the accuracy of the likelihood computation. We approximate the probability of genotype $y^{(i)}$ by drawing $L$ samples from each of the proposal distributions,

$$\Pr(Y = y^{(i)}) \approx \frac{1}{L} \sum_{l=1}^{L} \frac{\Pr(Y = y^{(i)} | x_l^{(i)}) \Pr(x_l^{(i)})}{Q(x_l^{(i)}, z_l^{(i)})}. \tag{8}$$

## Structure learning

Gerstung *et al.* [31] implemented a simulated annealing (SA) algorithm with a geometric annealing schedule to infer the network structure of the H-CBN model. However, as the size of the model increases, the poset search space increases rapidly (sequence A001035 in The On-Line Encyclopedia of Integer Sequences, https://oeis.org/A001035), and the standard SA algorithm is more prone to converge to local optima and to miss globally optimal or near-optimal solutions. Here, we incorporate an adaptive simulated annealing (ASA) algorithm [38] to improve the efficacy of the search. As in the standard SA algorithm [39], in each iteration, we

propose an update $P'$ of the current poset $P^{(k)}$ and accept the new poset with probability

$$\min\left(1, \exp\left(\frac{-[\ell_Y(\hat{\lambda}^{(k)}, \hat{\epsilon}^{(k)}, P^{(k)}) - \ell_Y(\hat{\lambda}', \hat{\epsilon}', P')]}{\Theta^{(k)}}\right)\right).$$

Conventionally, the temperature $\Theta$ is gradually reduced over iterations, initially allowing the system to explore a broad region of the search space, but ultimately moving exclusively towards solutions that improve the likelihood. In the ASA algorithm, the cooling schedule is adjusted according to the search progress, but following the same principle as before, i.e., gradually changing the temperature such that the system is able to converge [40, 41]. We have adopted the cooling schedule from Srivatsa *et al.* [42] as follows. For every interval consisting of $m$ consecutive iterations, we set the temperature $\Theta_m = \Theta_{m-1} \exp\left((0.5 - a_{m-1}^c)a_r\right)$, where $a_{m-1}$ is the observed acceptance rate of the previous interval, $a_r$ is a custom adaptation rate, and $c = \frac{-\log(2)}{\log a_{\text{ideal}}}$ is a scaling factor accounting for deviations from an optimal acceptance rate. Following the previous work [42], the optimal acceptance rate is set to $a_{\text{ideal}} = 1/p$, where $p$ is the number of mutations. Moreover, the adaptation rate $a_r$ is an additional free parameter enabling to further control the abruptness of temperature changes.

The optimization includes proposing a neighboring poset, which ultimately defines how we explore the space of posets. To this end, we implement three move types: (i) add or remove an edge, (ii) add an element to or remove an element from the cover relations while preserving all the remaining ones, and (iii) swap node labels. When adding an element to the cover relation, or equivalently an edge in the DAG, we discard proposed networks which are not transitively reduced or contain cycles.

## Implementation

We collectively refer to the implementation of the methods described in the previous sections as H-CBN2. It consists of the importance sampling schemes for parameter inference and the adaptive simulated annealing algorithm for structure learning of the H-CBN model. The code has been integrated into the MC-CBN R-package. We used C++ with OpenMP and the Boost libraries to ensure computational efficiency. We also employed the Vector Statistics component of the Intel Math Kernel Library (MKL) to efficiently generate random numbers.

## Statistical test for the comparison of CBN models

To compare two CBN models, we quantify differences in their posets using the Jaccard distance. The Jaccard distance between two sets is the complement of their Jaccard index, which is obtained by dividing the cardinality of the intersection by the cardinality of the union of the two sets.

Based on this notion of distance, we develop a permutation test to assess whether two given posets differ significantly from each other. Given two CBN models (e.g., estimated separately for HIV-1 subtypes B and C), we compute the Jaccard distance between the posets, $d_J$. The test quantifies how likely it is to observe the distance $d_J$ under the null hypothesis of both data sets having been generated by the same underlying poset. The alternative is that the two data sets have been generated by two different posets.

We compute the distribution of the test statistic $D_J$ under the null hypothesis as follows. We combine all genotypes from the considered groups and randomly split the data into two disjoint sets $S_1$ and $S_2$ with $N_1$ and $N_2$ genotypes, respectively, where $N_1$ and $N_2$ are the sizes of the two original data sets. That is, we permute the group labels of the genotypes. Then we apply H-CBN2 to infer the poset for $S_1$ and $S_2$ separately and compute their Jaccard distance. We

repeat this procedure $B$ times and construct the distribution of the test statistic $D_J$ under the null by aggregating the computed Jaccard distances (Fig 1). We assess how likely it is to observe a test statistic at least as extreme as $d_J$ under the null hypothesis by means of computing the associated p-value

$$\Pr(D_J > d_J \,|\, \mathcal{H}_0) = 1 - \hat{F}(d_J), \tag{9}$$

where $\hat{F}(d_J)$ is the empirical cumulative distribution function,

$$\hat{F}(d_J) = \frac{1}{B} \sum_{S_1, S_2 \,\in\, \mathcal{S}_B} \mathbb{I}(d_J(S_1, S_2) \le d_J), \tag{10}$$

where $\mathcal{S}_B = \{(S_{1_j}, S_{2_j}) : j = 1, \ldots, B\}$. For the comparisons, we choose a significance level of 5% and perform $B = 50$ permutations of the group labels.

The code for performing this distance-based test is available at https://github.com/cbg-ethz/H-CBN2-comparison-test.

## Simulated data sets

We simulate data sets with different error rates and various numbers of mutations ($p = 4, 8, 12, 16, 32, 64, 128$, and $256$). For each combination of the simulation settings, we generate 100 data sets with different rate parameters and different transitively reduced DAGs. We draw the rate parameters $\lambda_j$ uniformly at random from the interval $\left[\frac{\lambda_s}{3}, 3\lambda_s\right]$ ($\lambda_s = 1$). We also set the number of genotypes to $N = \max(50\,p, 1000)$, where the upper limit is motivated by the number of genotypes available in our application, i.e., comparing mutational pathways in different HIV-1 subtypes under lopinavir treatment (see next subsection).

For the assessment of the H-CBN2 importance sampling schemes, we choose error rates reflecting moderate to high Sanger sequencing error rates ($\epsilon = 0.01, 0.05$, and $0.10$) [43], which are challenging for parameter inference. To compare the H-CBN2 method to the predecessor method MC-CBN, we also simulated data with smaller error rates ($\epsilon = 0, 0.001$, in addition to 0.05 and 0.1) to assess the advantage of the latent error model compared to the simpler mixture error model under these milder conditions.

## Genotype data sets

We study lopinavir mutational networks in three data sets: (*i*) a cohort of 1064 South African patients living with HIV-1 subtype C retrieved from the Stanford HIV Drug Resistance Database (HIVDB, S1 File) [44, 45], (*ii*) a data set of 470 sequences of subtype B genotypes obtained from the HIVDB (S2 File) and the Swiss HIV Cohort Study (SHCS) [46, 47], and (*iii*) a data set of 755 sequences of subtype C genotypes obtained from the HIVDB excluding genotypes from South African patients contained in the first data set (S3 File).

The HIVDB is a publicly available database that systematically aggregates data from published studies about HIV drug resistance. The SHCS is a nation-wide, prospective observational study covering approximately 75% of all treated patients in Switzerland [47].

## Results

We first evaluate and compare the different importance sampling schemes implemented in H-CBN2 for the scalable inference of H-CBN models in a simulation study. We then apply the best performing H-CBN2 approach to investigate the accumulation of lopinavir resistance-associated mutations in HIV-1 in a large South African cohort. Finally, we compare the results

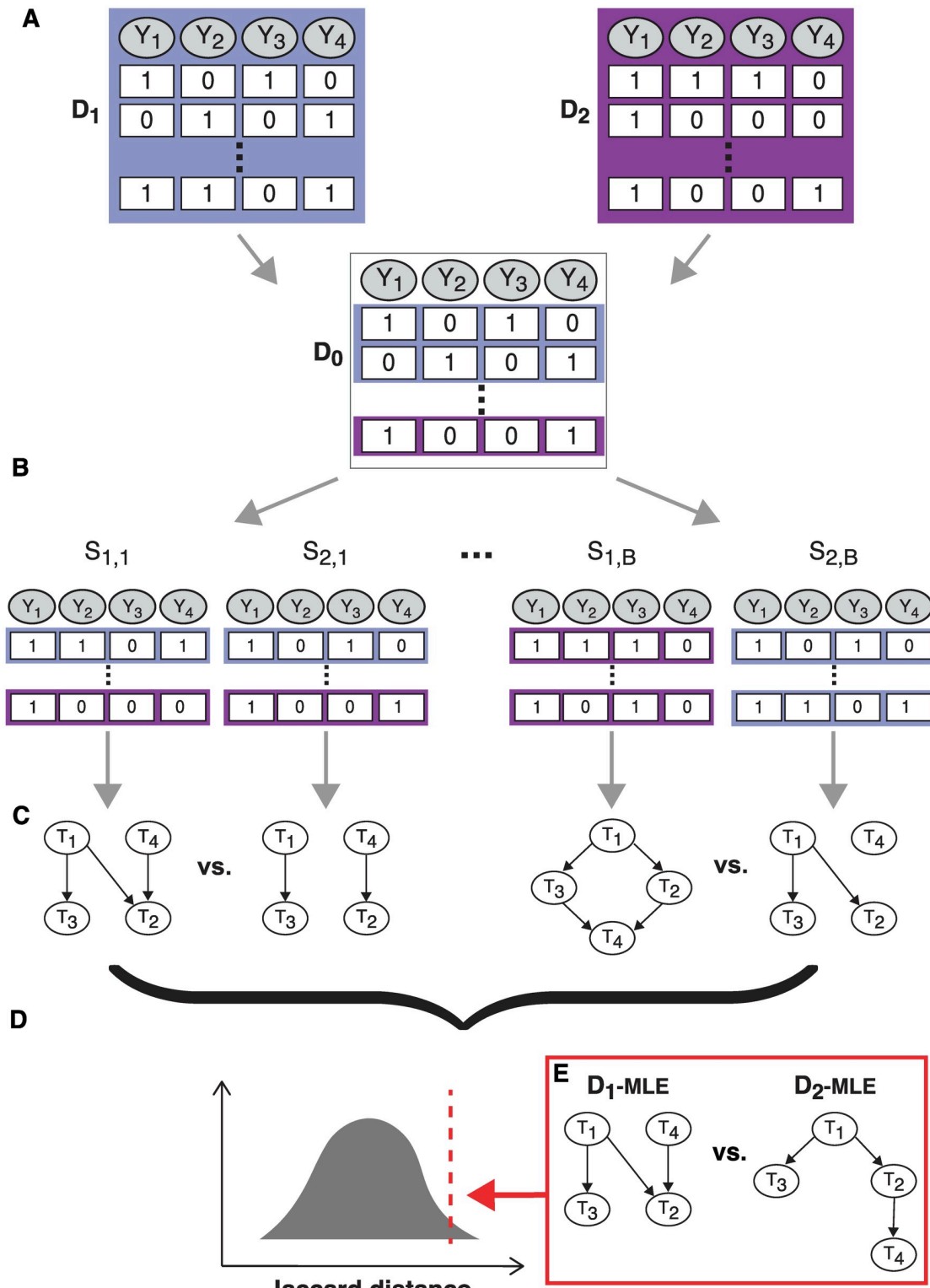

**Fig 1. Schematic representation of the comparison of CBN models. A** Data sets $D_1$ and $D_2$ consist of $N_1$ and $N_2$ genotypes, respectively, and, in this example, $p = 4$ mutations. We combined both data sets into a single one $D_0$ with $N_1 + N_2$ genotypes. **B** We randomly split data set $D_0$ into data sets $S_1$ and $S_2$ and we do so $B$ times. **C** For each data set, we apply the H-CBN2 approach to learn the structure of the network and for each pair, $S_1$ and $S_2$, we compute the Jaccard distance. **D** The empirical distribution of the test statistic is computed by aggregating the distances between pairs $S_1$ and $S_2$. **E** We compare the inferred CBN posets from original data sets $D_1$ and $D_2$ by computing the Jaccard distance and assess its significance.

for HIV-1 subtype C to a data set of HIV-1 subtype B genotypes derived from lopinavir-treated patients and obtained from the HIVDB and the SHCS.

## Simulation studies

**Assessment of importance sampling schemes on simulated data.** We assess the quality of approximating the probability of a genotype *y* by the H-CBN2 importance sampling schemes (Eq 8) and compare it to the exact solution. For a problem size relevant for our application ($p = 16$ mutations), we vary the number of samples, *L*, drawn from the proposal distribution and find, as expected, that the accuracy of the approximation improves with *L* (S1–S5 Figs). In most cases, we are able to accurately approximate the likelihood of the H-CBN model with $L = 1000$ (relative absolute error $\leq 0.02$, S2 Table).

We then assess the accuracy of parameter estimation using the H-CBN2 sampling schemes for various poset sizes and compare it to the H-CBN model [31]. We use a fixed number of samples drawn from the proposal distribution for each of the sampling schemes. We draw $L = 10$ samples for the Hamming 3-neighborhood sampling, $L = 100$ samples for the forward-pool sampling, and $L = 1000$ for the other sampling schemes. These choices are motivated by the preceding results on the quality of the log-likelihood approximation via importance sampling (S2 Table).

We evaluate the performance of the H-CBN2 sampling schemes based on the deviation from the true value of the estimated error rate $\hat{\epsilon}$ and rate parameters $\hat{\lambda}$. To summarize results for all the different rates $\hat{\lambda}_j$, we compute the relative (median) absolute error, which is given by $\frac{\text{median}(|\hat{\lambda}-\lambda|)}{\text{median}(\lambda)}$. Generally, we observe that for a known poset *P*, the estimation of the error rate and the rate parameters is accurate for small- and medium-sized posets (of up to about $p = 32$ mutations) under the evaluated conditions in terms of the sample size *N* and number *L* of samples drawn from the proposal distribution (Fig 2A and 2B and S7 Fig, see also S2 Text). In particular, the relative error in estimating the rate parameters λ increases more drastically for data sets with more than 32 mutations. This is likely due to the number of genotypes being limited to 1000, as well as the density of the network. In fact, mutations which depend on the occurrence of many predecessors are effectively only rarely encountered in the data sets as a consequence of the noise (S8 Fig). Therefore, there is often little or no evidence in the data to estimate the corresponding rate parameters. The estimates obtained by the Bernoulli sampling quickly deteriorate as the number of mutations increases, and for data sets with more than 64 mutations, the relative median error for the rate parameters is outside the range displayed in Fig 2A and 2B. This is because the fraction of incompatible genotypes increases with the number of mutations and it becomes less likely to sample candidate genotypes with non-zero weight (i.e., compatible with the poset, S4 Fig). Only for posets with up to 12 mutations, we can compare results to the H-CBN model. We find that most sampling schemes perform as well as the H-CBN in terms of the accuracy of the estimated parameters.

We also assess the run time performance of various sampling schemes implemented in H-CBN2 and compare to H-CBN (Fig 2C and S9 Fig); each run corresponds to 100 iterations of the MCEM or EM algorithm. We observe that H-CBN is faster than any of the H-CBN2 sampling schemes for posets with up to 6 mutations. Nonetheless, with the exception of the Hamming 3-neighborhood sampling scheme, we find an almost linear relationship between the number of mutations and the run time of the H-CBN2 sampling schemes, whereas the H-CBN run time grows exponentially with the number of mutations and is outperformed by H-CBN2 for $p \gtrsim 10$. We also observe that, for larger posets, the forward-pool sampling is slower than the standard forward sampling, because the size of the pool increases with the

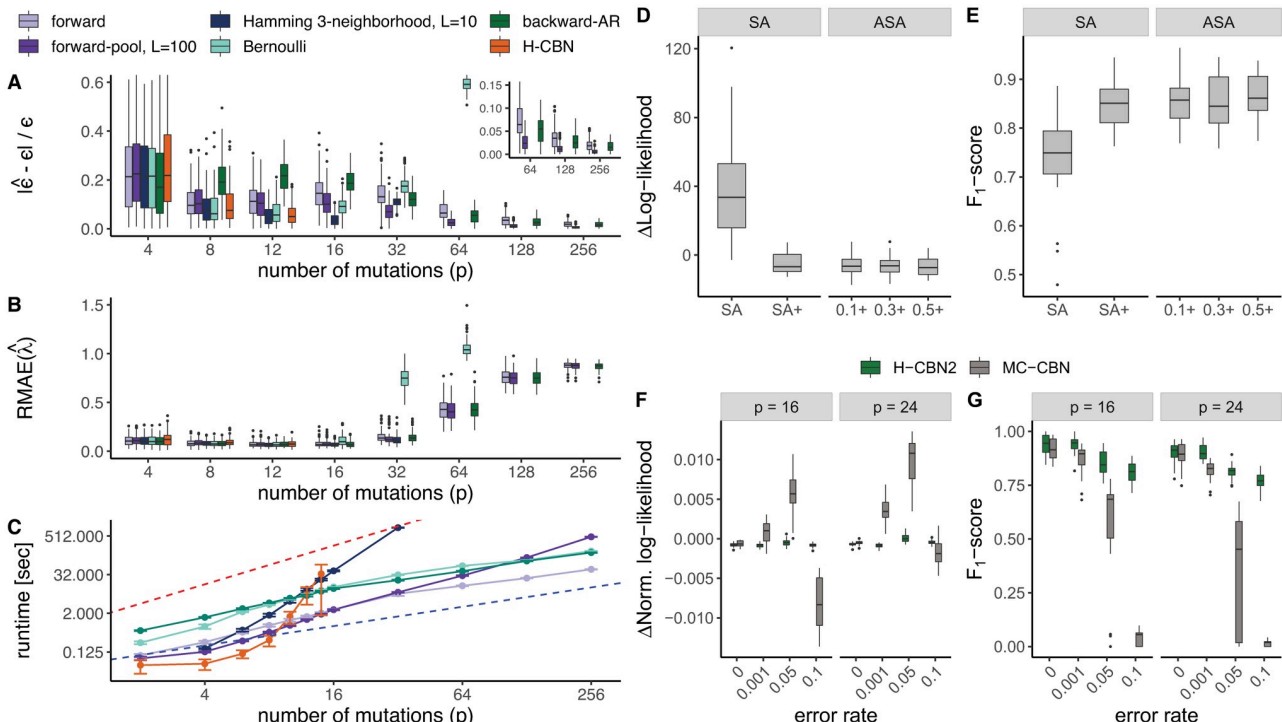

**Fig 2. Assessment of H-CBN2 on simulated data. A** Box plots of the difference between true ($\epsilon$) and estimated ($\hat{\epsilon}$) error rate (y-axis) for each of the evaluated poset sizes (x-axis). **B** Box plots of the relative median absolute error (RMAE; y-axis) of the estimated rate parameters $\hat{\lambda}$. **C** Average run time of the MCEM/EM step (y-axis, logarithmic scale) for different poset sizes (x-axis, logarithmic scale). The blue dotted line corresponds to linear scaling, whereas the red line corresponds to quadratic scaling. In panels A to C, different colors indicate different importance sampling schemes and we show results of 100 simulated data sets for each of combination of the simulation settings. The true error rate is $\epsilon = 0.05$, the number of samples drawn from the proposal distribution is set to $L = 1000$ unless specified otherwise and we run 100 iterations of the MCEM/EM algorithm. **D** Error in the estimation of the log-likelihood, $\ell(\lambda, \epsilon, P) - \ell(\hat{\lambda}, \hat{\epsilon}, \hat{P})$. **E** Box plots of $F_1$ scores for reconstructed network edges. In panels D and E, we show results of 20 different networks with 16 mutations and an error rate of 5%. We fix the ideal acceptance rate to $1/p$, and run 25,000 iterations of the simulated annealing algorithm. The initial temperature is set to $\Theta_0 = 50$ for all runs, and for adaptive simulated annealing, three adaptation rates are evaluated ($a_r = 0.1, 0.3, 0.5$). Comparison of H-CBN2 to MC-CBN methods in terms of **F** the difference in normalized log-likelihood and **G** $F_1$ scores for two poset sizes and various error rates. For the H-CBN2 results shown in panels F and G, we employ the ASA algorithm. SA: simulated annealing, ASA: adaptive simulated annealing, +: with additional new moves.

number of mutations; we set $K = pL$ to assure accurate parameter estimates (S2 Fig). As the number of mutations increases, the computation time of the Hamming distance becomes the limiting factor (Eq 6).

The forward sampling and the backward-AR sampling perform equally well in terms of accuracy of the estimated model parameters for small- and medium-sized posets, even when the number $L$ of samples drawn from the proposal distribution is set to 100 for the backward-AR sampling (S6(G) and S6(H) Fig). The run times of these sampling schemes with $L = 1000$ and $L = 100$, respectively, are also similar. The forward and the backward-AR sampling schemes thus enable performing parameter estimation for posets with more than 14 mutations and up to about 32 mutations. Since we do not observe any advantage in using the backward-AR sampling over the forward sampling, we choose the latter for the comparisons of mutational networks presented in this work.

**Assessment of the simulated annealing algorithm on simulated data.** So far, we assumed that the poset $P$ is known. In the following, we evaluate the performance of the H-CBN2 structure learning algorithm, which, in addition to adding or removing an edge, includes new moves to propose candidate posets, as well as an ASA schedule. We employ a

similar approach as before: (i) draw a transitive reduced DAG and parameters at random, (ii) generate a data set from the joint probability distribution of the model, and (iii) infer the network structure in addition the model parameters.

We first compare the accuracy of the estimated model parameters when the poset $P$ is also learned. We do not observe any manifest difference in the absolute error between the true and the estimated error rate (S10(A) Fig), but the relative absolute error of the rate parameters is marginally larger when the poset is learned in addition to the model parameters, as well as the absolute error of the log-likelihood (S10(B) and S10(C) Fig).

Next, we compare different SA strategies for structure learning. We observe a notable improvement in the log-likelihood of the reconstructed network after including the additional new moves (SA+) compared to a simulated annealing algorithm (SA) with only addition and removal of edges (Fig 2D). Incorporating, in addition, an adaptive annealing schedule yields similar performance to SA+. Similarly, the error in estimation of the model parameters also decreases mostly upon including the new moves (S11(A) and S11(B) Fig). We also compute the harmonic mean of precision and recall, i.e., the $F_1$ score, of reconstructing the elements of the cover relation and find a clear improvement of SA+ over SA (Fig 2E).

Finally, we investigate the influence of the annealing schedule hyper-parameters, such as the initial temperature and the adaptation rate (S10 Fig). In general, the performance of the ASA algorithm is not critically influenced by the choice of the annealing hyper-parameters. Moreover, the ASA algorithm is neither better nor worse than the SA+ algorithm, at least for the test cases with $p = 16$ mutations. Nevertheless, the ASA algorithm has the conceptual advantage of adjusting the temperature adaptively according to the system behaviour rather than using a fixed schedule and thus may be more reliable across unknown likelihood landscapes.

**Comparison with MC-CBN method.**   Comparisons of the CT-CBN, including the H-CBN, model to related models show that CT-CBNs oftentimes perform better in reconstructing the mutational networks for most of the evaluated metrics [32–34]. Here, we compare two recent CT-CBN methods, namely H-CBN2 and MC-CBN. We find that the gap between the log-likelihood of the data for the underlying and the learned models is, in general, smaller for H-CBN2 than for MC-CBN (Fig 2F) indicating a better fit. But for data sets with 10% error rate, overfitting attributed to the MC-CBN error model yields networks with no or only a few edges. These networks are highly dissimilar to the true structures as indicated by the low $F_1$ score for the reconstructed network edges (Fig 2G). More generally, for all positive error rates, H-CBN2 outperforms MC-CBN in learning the underlying DAG structure (Fig 2G). For error-free genotypes ($\epsilon = 0$), we find that both methods perform equally well, which confirms the validity of the H-CBN2 method given that the MC-CBN sampling scheme is most efficient in this limit.

## Comparison of drug resistance-associated mutational networks in different HIV-1 subtypes under lopinavir treatment

We analyze viral genotypes from a cohort of 1064 South African patients living with HIV-1 subtype C retrieved from the HIVDB. These patients were treated with lopinavir boosted with low-dosed ritonavir. We select a subset of 21 major protease inhibitor (PI) resistance mutations associated with lopinavir resistance and 15 non-polymorphic accessory mutations according to the HIVDB [45] (S12 Fig). We follow the convention of reporting mutations relative to the amino acid sequence of the HIV-1 subtype B reference strain HXB2. Among the selected loci, HIV-1 subtype C sequences typically differ at residue 89 from the subtype B

reference strain: instead of a leucine (L) a methionine (M) is frequently observed [48]. This naturally occurring polymorphism is found in 77.09% of the patient in our cohort.

We find 15 out of the 21 major PI mutations and 13 out of 15 non-polymorphic accessory mutations in the South African cohort. The remaining unobserved mutations include PI mutations G48M, I54A/L/M/T, and V82T, and non-polymorphic mutations L24F and A71I. We exclude polymorphisms commonly found in wild type subtype C viruses as they likely correspond to baseline mutations, but some of these are highly prevalent in the cohort—for instance, I93L (97.18%), M36I (86.95%), and K20R (34.18%). Among the 1064 genotypes, 911 are wild type for the selected loci and the maximum number of co-occurring mutations is eight.

In addition, we analyze two genotype-treatment data sets from the HIVDB corresponding to HIV-1 subtype B and C genotypes. For the latter, we exclude genotypes from South Africa that constitute the data set described above. All patients in these data sets were treated with lopinavir or lopinavir and ritonavir but not with another protease inhibitor. The data sets include 298 and 775 sequences of subtype B and C genotypes, respectively. Additionally, we consider 172 HIV-1 subtype B sequences of the SHCS derived from patients treated with lopinavir as the first PI. We jointly analyze all 470 subtype B genotypes to mitigate the small sample size.

We use H-CBN2 for analyzing and comparing the accumulation of resistance mutations in HIV-1 subtype B and subtype C under the selective pressure of lopinavir. We employ the forward sampling scheme to learn the partial order among mutations. The robustness of the network estimation is investigated by using 100 bootstrap samples and the consensus networks are shown in Figs 3A and 4 ($p = 20$ and $p = 18$, respectively). In the South African cohort (subtype C sequences), we identify a mutation at residue 82 in the protease as an early event. The initial substitution is likely to be V82A, as it is predominantly observed in the data set (S12 Fig). After this initial event, we find strong support for mutations at residues 10, 33, 46, 54 and 76 (Fig 3A). For subtype B, we find strong support for a mutation at residue 46 as an initial event (Fig 4). The inferred posets can explain previously observed mutation patterns, such as M46I+I54V alone or in combinations with L76V or V82A in subtype B [36], as well as M46I+I54V+V82A and L10F+M46I+I54V+L76V+V82A in subtype C [37].

At first glance, the subtype-specific H-CBN2 posets appear to be different. However, they also share many features. We find that they have 5 cover relations in common, namely, I54V $\prec$ L24I, I54V $\prec$ F53L, I54V $\prec$ G73S, I54V $\prec$ T74P, and I54V $\prec$ L89V. In addition, in both posets mutation at residue 82 precedes G73S and T74P, and mutation at residue 46 precedes K43T, F53L, T74P, and L89V either in a direct manner or through an intermediary event.

To assess whether the two H-CBN2 posets are significantly different beyond reconstruction uncertainty, we have developed a customized statistical test based on the Jaccard distance between the posets. The distance between the maximum likelihood posets (S13 and S14 Figs) is 0.802. To assess the significance of this result, we compare it to the empirical distribution of pairwise distances computed between reconstructed networks after randomly permuting the group labels (Fig 5). At a significance level of 5%, we reject the null hypothesis that the data sets stem from the same underlying poset (p-value $< 0.02$, Fig 5B), for $p = 18$ mutations. Similarly, we reject the null hypothesis while comparing subtype-specific CBN models for HIV-1 subtype B and C with $p = 11$ mutations (p-value $= 0.04$, Fig 5A). The smaller data sets are obtained by discarding mutations with marginal counts less or equal 5 in either of the two data sets.

As a negative control, we also compare the two H-CBN2 models for subtype C inferred from the South African cohort versus the remaining subtype C genotypes from the HIVDB (Fig 3). The consensus posets share 16 cover relations, namely, L10FR $\prec$ K43T, L10FR $\prec$

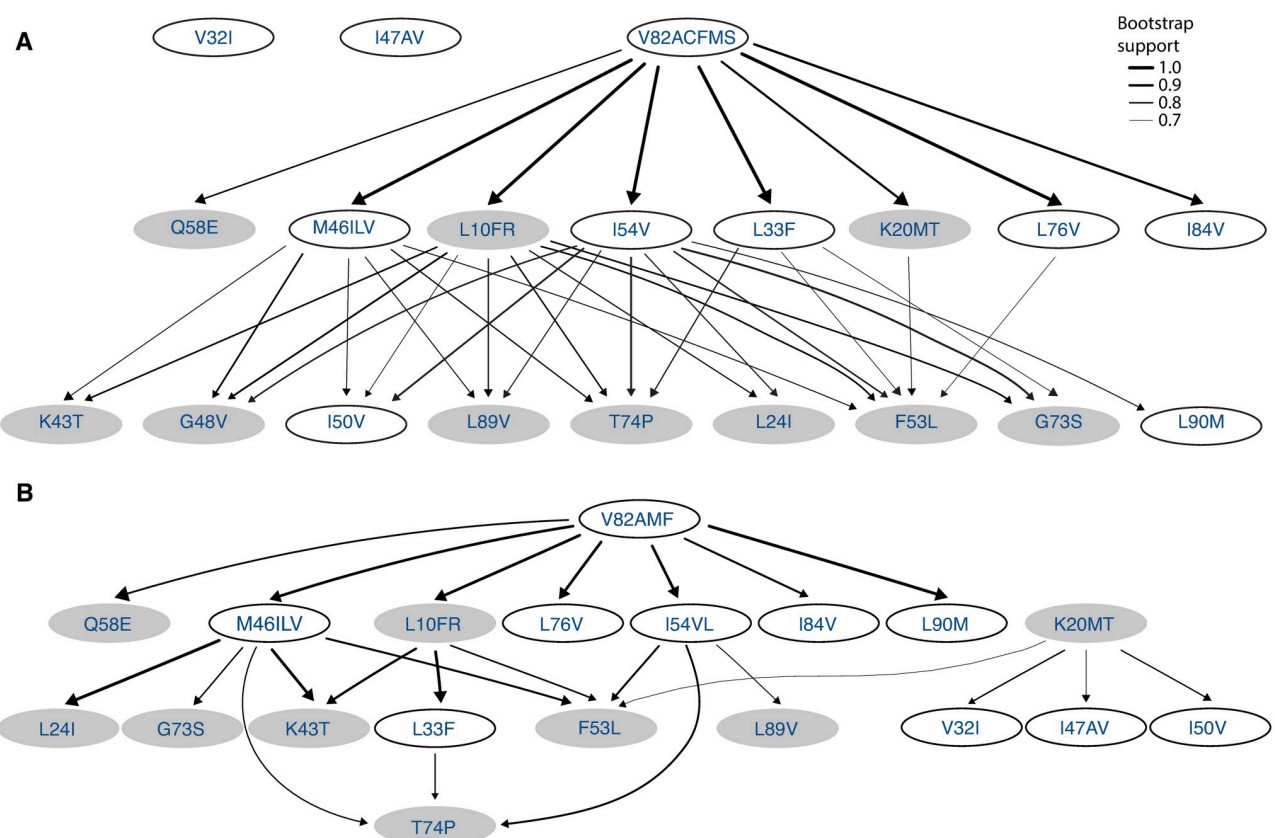

**Fig 3. Consensus posets for lopinavir resistance for two different HIV-1 subtype C data sets.** Shown are the consensus poset for **A** the South African cohort and **B** for the remaining HIV-1 subtype C sequences retrieved from the HIVDB. Nodes in the network correspond to amino acid changes in the HIV-1 protease, where mutations at the same locus are grouped together in one event. Only edges with a bootstrap support greater than 0.7 are shown and the edge thickness indicates the bootstrap support. Nodes with white background show residues with at least one major PI mutation.

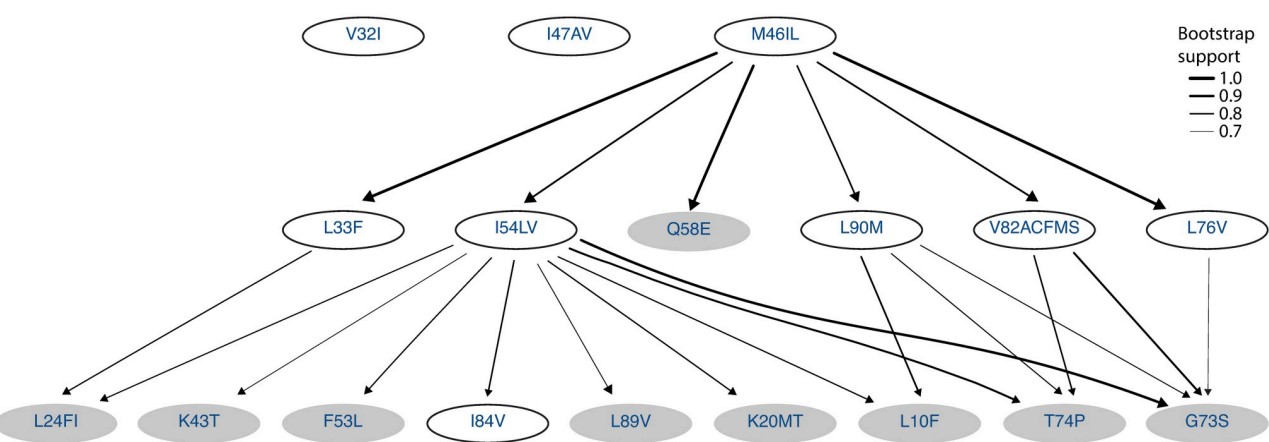

**Fig 4. Consensus poset for the accumulation of mutations in HIV-1 subtype B under lopinavir treatment.** The underlying data set contains 470 genotypes retrieved from the HIVDB and SHCS. Nodes in the network correspond to amino acid changes in the HIV-1 protease, and mutations at the same locus are grouped together. Edge labels indicate the bootstrap support, and we show only edges with a bootstrap support greater or equal to 0.7.

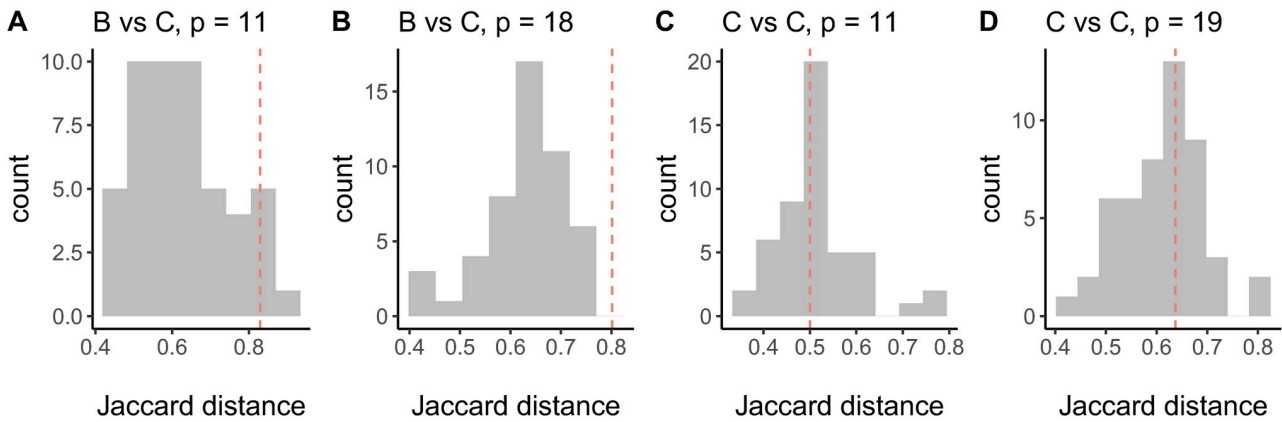

**Fig 5. Empirical null distribution of pairwise Jaccard distances estimated by permuting group labels.** Displayed are the histograms of Jaccard distances for the comparison of subtypes B and C for H-CBN2 posets with **A** 11 mutations and **B** 18 mutations, as well as the histograms of Jaccard distances for the comparison of two data sets for subtype C for H-CBN2 posets with **C** 11 mutations and **D** 19 mutations. Vertical dotted lines indicate the distance between the CBNs obtained from the observed data.

F53L, K20MT ≺ F53L, L33F ≺ T74P, M46ILV ≺ K43T, M46ILV ≺ F53L, M46ILV ≺ T74P, I54V/L ≺ F53L, I54V/L ≺ T74P, I54V/L ≺ L89V, V82AMF/CS ≺ L10FR, V82AMF/CS ≺ M46ILV, V82AMF/CS ≺ I54V/L, V82AMF/CS ≺ Q58E, V82AMF/CS ≺ L76V, and V82AMF/CS ≺ I84V. Moreover, in both posets mutation at residue 10 precedes T74P and mutation at residue 82 precedes L24I, L33F, K43T, F53L, G73S, T74P, L89V, and L90M. We also employ the aforementioned statistical test to compare posets with different number of mutations, namely $p = 19$ and $p = 11$ mutations. The larger poset size corresponds to all the mutated loci common in both data sets and the threshold on the marginal mutation counts for constructing the smaller data sets is set to 8 mutations. The Jaccard distance between these two H-CBN2 models is 0.637 and 0.5 for posets with $p = 19$ and $p = 11$ mutations, respectively. There is no evidence supporting that the posets learned from different data sets but the same subtype C are different (p-values 0.42 and 0.66, respectively; Fig 5C and 5D).

## Discussion

We have presented the H-CBN2 model and inference methods which are based on Monte Carlo sampling and enable us to consider a larger number of mutations. In simulation studies, we demonstrated that this method can be used to accurately estimate model parameters for up to about 32 mutations. For larger numbers of mutations, the sample sizes used in this work are insufficient to obtain accurate parameter estimates. To learn the graph, we proposed an extension of the simulated annealing algorithm, including additional move types that allow exploring the space of posets more efficiently. We validated the structure learning algorithm for 16 mutations which aligns with the numbers of mutations relevant for our application to HIV-1. Structure learning is, however, a hard problem and further improving the efficiency of this step might be worthwhile addressing in future research.

Even though there are descriptive analyses of subtype-specific PI mutation profiles [5, 36, 37, 48], to our knowledge, this study is the first comparative analysis of pathways of accumulating mutations over time in different HIV-1 subtypes. In addition to a more systematic approach to investigating mutation patterns, the number of observations in our study is greater than in any of the previous studies, which ranged from 88 to 165 patients. We applied the H-CBN2 approach to learn the partial temporal ordering of resistance mutations in HIV-1 subtypes B and C under the selective pressure of lopinavir. Our results indicate that despite

some similarities, for the considered numbers of mutations, the mutational networks differ significantly between the two subtypes. Moreover, we compared H-CBN2 posets for subtype C inferred from two independent data sets as a validation of the distance-based test and the outcome aligns with the expectation that there exists a single underlying poset explaining both data sets better than two distinct posets.

In our analysis, we included major PI mutations associated with lopinavir resistance and non-polymorphic accessory mutations. Although some polymorphisms, in combination with PI resistance mutations, are associated with an increase in viral fitness [49], these are also highly prevalent in treatment-naïve patients, especially in non-B subtypes [50–52]. Therefore, despite observing polymorphisms with relatively high prevalence, we did not include these mutations in our study. We also found more than one PI-associated mutation in only about 14% and 16% of the patients in the South African cohort (subtype C) and the subtype B data set, respectively. The absence of resistance mutations in the protease gene has been repeatedly observed at virological failure, even in the absence of reverse transcriptase inhibitors [36, 53– 56]. In addition to poor adherence to treatment [57, 58], there may be other reasons for observing a low percentage of patients harboring PI resistance mutations, and some of them are listed below. First, the genetic barrier to lopinavir resistance appears to be high. Barber *et al*. [36] have suggested that PI resistance mutations are more likely to accumulate under prolonged virological failure. Second, there is increasing evidence that mutations in the *gag* gene play a role in decreasing susceptibility to protease inhibitors by, e.g., inhibiting the proteolytic cleavages necessary for protein maturation [23, 59, 60]. Virions with immature particles may not adequately complete cell entry or reverse transcription [60]. Third, although there is no clear consensus on the clinical relevance of minority variants, resistance mutations may exist in the intra-host virus population at frequency below the detection threshold. Mutations are typically detected by Sanger sequencing-based methods, while next-generation sequencing methods could improve upon the sensitivity of detecting mutations [61, 62]. Higher-resolution data, including multiple longitudinal samples from the same patient, offer a different perspective on studying evolution of drug resistance. In this case, phylogenetic models are well-suited to model the evolutionary process within individual hosts. We underline that progression models, such as CBNs, are designed for cross-sectional consensus sequences collected from different patients under similar conditions to model the accumulation of mutations under specific evolutionary constraints, such as the same selective drug pressure. In contrast to phylogenetic methods, progression models do not seek to reconstruct the joint evolutionary history of the observations, but rather aim at capturing reproducible features of the evolutionary process in different hosts.

We have made several simplifying assumptions in our model. For example, we treat different amino acid substitutions at the same locus as indistinguishable events. This is because when observing a specific substitution, we do not know which other mutations at that locus might have led to the current state. Also, the type of dependencies that CBNs can model is limited. Certain fitness landscapes and the epistatic interactions they encode may give rise to genotype probabilities that CBNs can not represent. While some forms of both positive and negative interactions can be captured, the effects of other epistatic interactions (e.g., reciprocal sign epistasis) can not be represented by CT-CBNs [33, 34]. Allowing more complex interactions, would require to account for different transitions between genotypes with a concomitant increase in the number of model parameters. We consider the CT-CBN models to exhibit a good trade-off between low model complexity and goodness of fit to the observed data.

The comparison of the cross-sectional data sets is challenging due to the existence of several confounders. First, the data are gathered from various sources, which entails potential differences in HIV surveillance and clinical monitoring protocols. Moreover, observations come

from distinct geographical locations, which implies, e.g., differences in socio-demographic aspects and health-care standards. Lastly, therapeutic strategies tend to differ between developed and developing countries, and there is a limited number of observations of various subtypes undergoing the same therapy. In the present study, the number of observations in the subtype B data set is approximately half of the observations available for subtype C. Such an imbalance poses additional challenges for the CBN comparisons. The spread of the empirical distribution of Jaccard distances might be wider for imbalanced data sets, which could result in an apparent increase in false negatives. But rather than a shortcoming of the distance-based test, small sample size generally lead to reduced accuracy of the parameter estimates, including the network structure.

In summary, the inferred CBN models provide insights into the evolution of drug resistance in HIV-1 subtype C infections and enable comparisons with other subtypes, as demonstrated for subtype B. Moreover, the methods proposed in this work can be applied to investigate subtype-associated differences pertaining to HIV-1 drug resistance, but more generally the methodology can be adopted for comparing any two groups in the context of other evolutionary process.

## Supporting information

**S1 Text. Additional notes on the model and parameter estimation.**
(PDF)

**S2 Text. Additional details on the assessment of importance sampling schemes on simulated data.**
(PDF)

**S1 Fig. Assessment of the forward sampling.** Probability of the observed genotype estimated by using the forward sampling scheme $\widetilde{\Pr}(Y = y)$ (y-axis, Eq 8) vs. the exact solution $\Pr(Y = y)$ (x-axis). The data set consists of $N = 800$ genotypes with $p = 16$ mutations and an error rate of 5%. Results are obtained by drawing **A** $L = 10$, **B** $L = 100$, and **C** $L = 1000$ samples from the proposal distribution.
(PDF)

**S2 Fig. Assessment of the forward-pool sampling.** Probability of the observed genotype estimated by using the forward-pool sampling scheme $\widetilde{\Pr}(Y = y)$ (y-axis, Eq 8) vs. the exact solution $\Pr(Y = y)$ (x-axis). The data set consists of $N = 800$ genotypes with $p = 16$ mutations and an error rate of 5%. First, we evaluate the impact of the size of the initial pool on the accuracy of the approximations. We show results for pools consisting of **A, D** $K = 200$, **B, E** $K = 800$, and **C, F** $K = 1600$ samples, while the number of samples drawn from the proposal distribution is set to either **A-C** $L = 10$ or **D-F** $L = 100$. Next, we evaluate the impact of the number of samples drawn from the proposal distribution (**G** $L = 10$; **H** $L = 100$; **I** $L = 1000$), while the size of the initial pool is kept constant at $K = 2000$ samples. We observe that the accuracy of the computation improves primarily as the size of the initial pool increases. By default, the size of the initial pool of waiting times is set to $K = p \times L$.
(PDF)

**S3 Fig. Assessment of the Hamming $k$-neighborhood sampling.** Probability of the observed genotype estimated by using the Hamming $k$-neighborhood sampling scheme $\widetilde{\Pr}(Y = y)$ (y-axis, Eq 8) vs. the exact solution $\Pr(Y = y)$ (x-axis). The data set consists of $N = 800$ genotypes with $p = 16$ mutations and an error rate of 5%. Results are shown for **A-C** a neighborhood including the leading and the first-order terms ($k = 1$), **D-F** a neighborhood including the

leading, the first-order, and the second-order terms ($k = 2$), and **G-I** a neighborhood including the leading, the first-order, the second-order, and the third-order terms ($k = 3$). In this case, the value of $L$ indicates the number of waiting time vectors sampled per genotype in the neighborhood.
(PDF)

**S4 Fig. Assessment of the Bernoulli sampling.** Probability of the observed genotype estimated by using the Bernoulli sampling scheme $\widetilde{\Pr}(Y = y)$ (y-axis, Eq 8) vs. the exact solution $\Pr(Y = y)$ (x-axis). The data set consists of $N = 800$ genotypes with $p = 16$ mutations and an error rate of 5%. Results are obtained by drawing **A** $L = 10$, **B** $L = 100$, and **C** $L = 1000$ samples from the proposal distribution. In the lower panel, we show the number of samples compatible with the poset $L_{\text{compatible}}$ per genotype for **D** $L = 10$, **E** $L = 100$, and **F** $L = 1000$ samples drawn from the proposal distribution.
(PDF)

**S5 Fig. Assessment of the backward-AR sampling.** Probability of the observed genotype estimated by using the backward-AR sampling scheme $\widetilde{\Pr}(Y = y)$ (y-axis, Eq 8) vs. the exact solution $\Pr(Y = y)$ (x-axis). The data set consists of $N = 800$ genotypes with $p = 16$ mutations and an error rate of 5%. Results are obtained by drawing **A** $L = 10$, **B** $L = 100$, and **C** $L = 1000$ samples from the proposal distribution.
(PDF)

**S6 Fig. Assessment of the parameter estimation for various numbers of mutations and various numbers of samples drawn from the proposal distribution.** Box plots of the absolute error in estimating the error rate $\hat{\epsilon}$ for the true poset $P$ by using **A** the forward sampling, **D** the Hamming $k$-neighborhood sampling, and **G** the Bernoulli or the backward-AR sampling. Box plots of the relative median absolute error (RMAE) for the estimated rate parameters $\hat{\lambda}$ by using **B** the forward sampling, **E** the Hamming $k$-neighborhood sampling, and **H** the Bernoulli or the backward-AR sampling. Average run times over simulated data sets for **C** the forward sampling, **F** the Hamming $k$-neighborhood sampling, and **I** the Bernoulli or the backward-AR sampling. Results correspond to 100 simulated data sets for each of the number of mutations and 100 iterations of the MCEM algorithm. The number of samples drawn from the proposal distribution is $L = 10, 100, 1000$, as shown in the corresponding legend. The sample size is $N = \min(50\,p, 1000)$ and the true error rate is $\epsilon = 0.05$.
(PDF)

**S7 Fig. Assessment of parameter estimation for various numbers of mutations, error rates, and posets.** **A** Box plots of the relative difference between true ($\epsilon$) and estimated ($\hat{\epsilon}$) error rate (y-axis) for 100 simulated data sets for each of the evaluated model sizes (x-axis). **B** Box plots of the relative median error (RME; y-axis) of the estimated rate parameters $\hat{\lambda}$. The relative (median) error is given by $\frac{\text{median}(\hat{\lambda} - \lambda)}{\text{median}(\lambda)}$. Different colors indicate different importance sampling schemes. The sample size is $N = \min(50\,p, 1000)$ and the number of samples drawn from the proposal distribution is set to $L = 1000$ unless specified otherwise. We run 100 iterations of the Monte Carlo EM algorithm.
(PDF)

**S8 Fig. Understanding the source of bias in estimating rate parameters.** Box plots of **A** the relative median error (RME) and **B** the relative median absolute error (RMAE) for the estimated rate parameters $\hat{\lambda}$, while varying the numbers of order constraints (S1 Table) and mutations, as well as varying the number of simulated genotypes as indicated in this figure legend.

**C** Percentage of mutations marginally observed in the true underlying genotypes. For these mutations, there is evidence in the data to estimate the corresponding rate parameters. Each box plot shows results for 100 different simulated data sets. The number of order constrains, or equivalently, the number of edges in the simulated networks is depicted as graph density, were 0 means independent mutations and 1 corresponds to a linear chain.
(PDF)

**S9 Fig. Average run time of the MCEM step (y-axis, logarithmic scale) using various sampling schemes for different poset sizes (x-axis, logarithmic scale).** We also show the run times of the H-CBN method for posets with up to 14 mutations. The benchmark is conducted on 100 different data sets per poset size, and the number of EM iterations is set to 100. The blue dotted line corresponds to linear scaling, whereas the red line corresponds to quadratic scaling. We conduct the benchmark on two 12-core Intel Xeon E5–2680 v3 processors (2.5 GHz).
(PDF)

**S10 Fig. Evaluation of the simulated annealing algorithm for various initial temperatures ($\Theta_0 = 10, 30, 50$) and adaptation rates ($a_r = 0.1, 0.3, 0.5$).** We show box plots corresponding to 20 different transitively reduced DAGs with 16 mutations and for an error rate of 5%. Gray box plots correspond to results of the MCEM algorithm for the true poset. We use the forward sampling scheme with $L = 1000$ samples. For learning the poset, we fix the ideal acceptance rate to $1/p = 0.0625$ and run 25000 simulated annealing iterations. P: true poset, SA: simulated annealing, +: with additional new moves.
(PDF)

**S11 Fig. Evaluation of the adaptive simulated annealing algorithm on simulated data. A** Absolute error in estimating the error rate parameter $\hat{\epsilon}$. **B** Relative median absolute error (RMAE) of the estimated rate parameters $\hat{\lambda}$. **C** Jaccard distance computed on the cover relation sets for the true and estimated poset. We show box plots corresponding to 20 different transitively reduced DAGs for simulated data sets with 16 mutations and an error rate of 5%. We use the forward sampling scheme with $L = 1000$ samples drawn from the proposal distribution. We fix the ideal acceptance rate to $1/p = 0.0625$ and run 25000 iterations of the simulated annealing algorithm. The initial temperature is set to $\Theta_0 = 50$ for all runs and for the adaptive simulated annealing three adaptation rates are evaluated ($a_r = 0.1, 0.3, 0.5$). SA: simulated annealing, ASA: adaptive simulated annealing, +: with additional new moves.
(PDF)

**S12 Fig. Marginal mutation frequencies observed in HIV-1 subtype B and subtype C populations under lopinavir treatment. A** Data collected from 1064 patients from South Africa (HIV-1 subtype C). **B** HIV-1 subtype C genotypes retrieved from the HIVDB, excluding genotypes in data set A. **C** HIV-1 subtype B genotypes retrieved from the HIVDB. **D** Data obtained from the SHCS corresponding to subtype B genotypes. Major protease inhibitor resistance mutations are shown in black.
(PDF)

**S13 Fig. Maximum likelihood poset for lopinavir resistance in the South African cohort.** Nodes in the network correspond to amino acid changes in the protease, with mutations at the same locus grouped together. Mutations G48V and I50V are excluded for the comparison of H-CBN2 models, as they are not observed in the subtype B data set.
(PDF)

**S14 Fig. Maximum likelihood poset for lopinavir resistance in HIV-1 subtype B.** Nodes in the network correspond to amino acid changes in the protease, with mutations at the same locus grouped together. Data sources: the HIVDB and the SHCS.
(PDF)

**S1 Table. Average (range) number of edges in simulated data sets.**
(PDF)

**S2 Table. Relative error in approximating the log-likelihood via importance sampling.** The relative error is computed by dividing the absolute error by the absolute value of the true log-likelihood. We note that for the forward-pool sampling, the relative absolute error of the log-likelihood depends almost exclusively on the the size of the initial pool $K$ for $L \geq 10$. Similarly, the approximation accuracy of the Hamming $k$-neighborhood sampling is primarily determined by the extent $k$ of the considered neighborhood.
(PDF)

**S1 File. Patient identifiers of the South African cohort as retrieved from the HIVDB.**
(CSV)

**S2 File. Patient identifiers of HIV-1 subtype B genotype sequences retrieved from the HIVDB.**
(CSV)

**S3 File. Patient identifiers of HIV-1 subtype C genotype sequences retrieved from the HIVDB and excluding genotypes from South African cohort.**
(CSV)

## Acknowledgments

The authors thank Lisa Lamberti for critical reading, Marek Pikulski for valuable discussions, and Huyen Nguyen for providing data sets from the SHCS. The authors thank the patients who participate in the Swiss HIV Cohort Study (SHCS); the physicians and study nurses, for excellent patient care; the resistance laboratories for high-quality genotyping drug resistance testing; SmartGene (Zug, Switzerland) for technical support; Alexandra Scherrer, Susanne Wild, Anna Traytel from the SHCS data center for data management; and Danièle Perraudin and Mirjam Minichiello for administration. The members of the SHCS include the following: Anagnostopoulos A, Battegay M, Bernasconi E, Böni J, Braun DL, Bucher HC, Calmy A, Cavassini M, Ciuffi A, Dollenmaier G, Egger M, Elzi L, Fehr J, Fellay J, Furrer H (Chairman of the Clinical and Laboratory Committee), Fux CA, Günthard HF (President of the SHCS), Haerry D (deputy of "Positive Council"), Hasse B, Hirsch HH, Hoffmann M, Hösli I, Huber M, Kahlert C, Kaiser L, Keiser O, Klimkait T, Kouyos RD, Kovari H, Ledergerber B, Martinetti G, Martinez de Tejada B, Marzolini C, Metzner KJ, Müller N, Nicca D, Paioni P, Pantaleo G, Perreau M, Rauch A (Chairman of the Scientific Board), Rudin C (Chairman of the Mother and Child Substudy), Scherrer AU (Head of Data Centre), Schmid P, Speck R, Stöckle M, Tarr P, Trkola A, Vernazza P, Wandeler G, Weber R, Yerly S.

## Author Contributions

**Conceptualization:** Susana Posada-Céspedes, Gert Van Zyl, Hesam Montazeri, Niko Beerenwinkel.

**Data curation:** Soo-Yon Rhee.

**Formal analysis:** Susana Posada-Céspedes.

**Funding acquisition:** Roger Kouyos, Huldrych F. Günthard, Niko Beerenwinkel.

**Methodology:** Susana Posada-Céspedes, Hesam Montazeri, Jack Kuipers, Niko Beerenwinkel.

**Resources:** Roger Kouyos, Huldrych F. Günthard.

**Software:** Susana Posada-Céspedes.

**Supervision:** Niko Beerenwinkel.

**Writing – original draft:** Susana Posada-Céspedes, Niko Beerenwinkel.

**Writing – review & editing:** Susana Posada-Céspedes, Gert Van Zyl, Hesam Montazeri, Jack Kuipers, Soo-Yon Rhee, Roger Kouyos, Huldrych F. Günthard, Niko Beerenwinkel.

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
