## [Decision Letter · Decision Letter 0]

22 Feb 2021

Dear Prof. Beerenwinkel,

Thank you very much for submitting your manuscript "Comparing mutational pathways to lopinavir resistance in HIV-1 subtypes B versus C" for consideration at PLOS Computational Biology.

We apologize for the unusually long period during which this manuscript was under review; we had a very difficult time securing reviewer commitments. Thankfully, the two experts who reviewed your manuscript offered well reasoned and detailed critiques, with clear requests for changes/improvements.

As with all papers reviewed by the journal, your manuscript was reviewed by members of the editorial board and by several independent reviewers. In light of the reviews (below this email), we would like to invite the resubmission of a significantly-revised version that takes into account the reviewers' comments.

We cannot make any decision about publication until we have seen the revised manuscript and your response to the reviewers' comments. Your revised manuscript is also likely to be sent to reviewers for further evaluation.

Sincerely,

Sergei L. Kosakovsky Pond, PhD

Associate Editor

PLOS Computational Biology

Thomas Leitner

Deputy Editor

PLOS Computational Biology

Reviewer's Responses to Questions

**Comments to the Authors:**

Reviewer #1: In the article, the authors present an extension of the Hidden Conjunctive Bayesian Network framework (H-CBN) for the inference of mutational graphs from cross sectional data (previously developed by some of the authors), aimed at improving the overall scalability, by proposing different sampling schemes and a new adaptive simulating annealing strategy.

They apply their method (named H-CBN2) to infer the mutational/evolutionary pathways related to lopinavir resistance in HIV-1 subtypes B and C and they compare the models, finding significant differences.

As a first general comment, the article is technically sound, even if the presentation of goals and assumptions is not always clear and effective.

On the one hand, the core of the article is the presentation of an extension of an already existing computational approach, aimed at addressing scalability issues (most of the main text and the whole SM), and this would plainly position the article within the Methods category.

On the other hand, starting from the title, the abstract and the author summary, the declared goal and focus of the manuscript seem to be exquisitely application/translational.

I would strongly suggest to choose either a methodological or a translational "cut" and to revise the overall organization accordingly and in a consistent fashion, starting from the title, abstract, summary, proportion of text in the sections, etc.

More importantly, the manuscript presents important shortcomings from both the methodological and the translational perspectives.

From the methodological perspective, a more comprehensive comparison against competing methods is surely missing, since the authors only compare their results with the model of H-CBN, which is also the inspiration/source of their approach.

There is plenty of methods to reconstruct mutational trees/graphs (which the authors themselves cite) and at least some of the most recent/used should be included in the comparison.

Furthermore, a comparison with phylodynamics approaches is missing, see, e.g. [Kühnert, D., Kouyos, R., Shirreff, G., Pečerska, J., Scherrer, A. U., Böni, J., ... & Stadler, T. (2018). Quantifying the fitness cost of HIV-1 drug resistance mutations through phylodynamics. PLoS pathogens, 14(2), e1006895]. This article is not even cited, despite being applied to one of the datasets employed in this work.

How do the proposed approach compares against standard phylodynamic approaches?

Would not it be more sound to infer a phylogeny of samples instead than a mutational graph, attaching the mutations to the phylogenetic model a posteriori? The authors should address this important concern.

An further general methodological concern regards the effectiveness of population models inferred from cross-sectional data, especially in presence of high or unknown levels of genomic heterogeneity, as typically observed in cancer evolution scenarios, which is the field within which H-CBNs were originally developed.

In this regard, in a recent work some of the authors introduced the concept of predictability, which roughly measures the repeatability of the evolutionary patterns of a given mutational graph [Hosseini, S. R., Diaz-Uriarte, R., Markowetz, F., & Beerenwinkel, N. (2019). Estimating the predictability of cancer evolution. Bioinformatics, 35(14), i389-i397].

This aspect must be evaluated if one aims at delivering experimental hypotheses with translational relevance and could help in casting a light on the intrinsic heterogeneity of the generative process underlying resistance in HIV.

Moreover, the suitability of conjunctive models has longly been questioned, for instance with respect to mutual exclusivity events (e.g., negative epistasis) or loss of mutations.

These aspect should be considered, particularly when considering a significantly high number of genomic events/mutations, which is the scenario that the authors aim to address by improving the scalability w.r.t. standard H-CBNs.

Besides, in [PLoS One. 2010 Jul 7;5(7):e11345. doi: 10.1371/journal.pone.0011345.] back mutations are indeed observed and "resistant virus disappeared rapidly after treatment interruption and was undetectable as early as after 3 months".

How do the authors can assess the robustness of their results, which seem to be highly dependent on sampling/sequencing timing?

In addition, how do the proposed approach accounts for positive epistasis, which is significantly present in HIV evolution (see, e.g., Bonhoeffer, Sebastian, et al. "Evidence for positive epistasis in HIV-1." Science 306.5701 (2004): 1547-1550.9)?

For instance, in [Zhang, Tian-hao, et al. "Predominance of positive epistasis among drug resistance-associated mutations in HIV-1 protease." PLoS genetics 16.10 (2020): e1009009] the authors claim: "We found that although resistance associated mutations greatly reduce replication fitness, they interact positively to alleviate the mutational load. These genetic interactions, termed epistasis, increase the ruggedness along the evolution paths, restricting resistance associated mutations from reversal."

In the same paper, the authors find at least two mutations (L10F and M46) with positive epistasis, which is in apparent contrast to the model presented in Figure 6 of the current manuscript, in which such mutations are located in independent branches.

From the application perspective, the presentation of the translational results derived from the models is quite poor and misses many important relevant aspects in HIV resistance analysis. More detail follows.

First, the article is completely missing the key issue of intra-host genomic variability.

For instance, the concept of quasispecies is not even mentioned and this is serious flaw, considering that the declared goal is to investigate resistance in viral evolution (see, for instance, the many thorough works by Esteban Domingo).

Quasispecies genetic "swarms" are supposed to underlie most of the adaptive potential of viruses to the immune system response or to antiviral therapies, also for HIV [see, e.g., Najera, Isabel, et al. "Pol gene quasispecies of human immunodeficiency virus: mutations associated with drug resistance in virus from patients undergoing no drug therapy." Journal of virology 69.1 (1995): 23-31.; Boutwell, C. L. et al. Todd M. Viral Evolution and Escape during Acute HIV‐1 Infection. The Journal of Infectious Diseases 202, S309–S314, https://doi.org/10.1086/655653 (2010). Hedskog, Charlotte, et al. "Dynamics of HIV-1 quasispecies during antiviral treatment dissected using ultra-deep pyrosequencing." PloS one 5.7 (2010): e11345. Vandenhende, M. A. et al. Prevalence and evolution of low frequency HIV drug resistance mutations detected by ultra deep sequencing in patients experiencing first line antiretroviral therapy failure. Plos One 9, e86771 (2014), Da Wei Huang, Castle Raley, et al. "Towards better precision medicine: PacBio single-molecule long reads resolve the interpretation of HIV drug resistant mutation profiles at explicit quasispecies (haplotype) level." Journal of data mining in genomics & proteomics 7.1 (2016).]

It is clear that the resolution of consensus sequences does not allow to evaluate the presence of low frequency variants or to deconvolve the quasispecies composition (as the author admit in the Discussion), yet this issue might significantly impact the overall translational relevance of this work and should be explicitly addressed.

Moreover, the authors state "However, when viral replication is inadequately suppressed, the virus may acquire mutations 6 which confer resistance to cART,".

Despite being a complex topic, it was hypothesized that treatment failure might be instead likely caused by the preexistence of resistant mutants (see, e.g., Proc Natl Acad Sci U S A. 2000 Jul 5; 97(14).

How can the H-CBN2 approach be effective in addressing the existence of pre-existing mutations, for instance if present in a subset of samples of the cohort?

The authors also claim: "A better understanding of the underlying evolutionary process leading to resistance is believed to be crucial for predicting therapy outcome and designing effective therapy sequences ".

I might have missed some detail, but if the therapy underlies the emergence of resistance-related mutations, the model cannot be used for predictions, but only for a posteriori explanations/descriptions.

Minor

Some sentences are involuted.

For instance, in the abstract "the maximum likelihood mutational networks for subtypes B and C share only 7 edges (Jaccard distance 0.802) and imply many different evolutionary

pathways.

What is an edge? How many edges are present in the models? Why reporting the Jaccard distance?

How were the error rates of the simulated datasets chosen?

abstract: Antiretoviral  Antiretroviral

Fig 1A is a very minor modification of Fig 1B of the article "Quantifying cancer progression with conjunctive Bayesian networks" and which should be mentioned.

Reviewer #2: The paper describes a general method for inference of mutation accumulation rates and orders of their appearance from genomic data. The authors propose a maximal likelihood-based graphical model for inference of evolutionary parameters of interest. The most general version of the model consists of exponential number of terms and variables and thus is computationally intractable; therefore, the paper proposes a mixed approach combining Monte Carlo Sampling and Expectation Maximization likelihood optimization for numerical parameter estimation and simulated annealing-based local search for graphical model inference. The proposed method is applied to the analysis of the emergence of lopinavir resistance mutations in HIV.

Overall, the proposed general modelling framework is elegant and may be of interest to the computational genomics community. However, I have some concerns about presentation, proposed approach and its validation.

1) The title is somewhat misleading. Judging by the way the manuscript is written, its major subject is the methodology rather than applications. The method is rather general and has nothing specifically related to viral drug and therapy resistance. As a matter of fact, it seems like it can be applied to any kind of viral or cancer genomic data.

2) Abstract claims that the proposed method can handle a large number of resistance mutations. However, the validation results on simulated data does not support this claim. Possibly it is meant that the method still can handle more mutations than its competitors – in that case it should be emphasized.

3) The motivation part should be improved. Specifically, it is claimed that the major novelty of the study is the comparison of mutational pathways between different HIV subtypes. However, it is not entirely clear, why such comparison cannot be made using existing methods, and why development of a new method facilitates such comparison.

4) The relation between the proposed approach and standard phylogenetic methods is mentioned only in passing. However, I feel that this relation needs to be discussed in greater detail. It is unclear why mutational pathways and rates cannot be inferred using classical maximum likelihood or Bayesian phylodynamics methods with relaxed molecular clock and ancestral states reconstruction.

5) The error model used in the study seems to be too simplistic. The developed method is applied to genotypes produced by different sequencing experiments, sequencing platforms, and laboratories. Therefore, the assumption of the same error rate for all genotypes is unrealistic.

6) The validation on simulated data is quite limited. The method is basically compared with itself under different settings. It would be desirable to add comparisons with existing methods or at least with some alternative approaches.

7) In simulation study results, it is unclear why the absolute error is reported for genotype error rate and the relative error – for mutation rate. I believe the same relative error should be reported for all parameters.

8) It is unclear why the results for simulated annealing algorithm are reported only for 16 mutations. If the results for larger datasets are poor (as Discussion seems to confirm), they still need to be reported. If this is the case, then is it related to optimality of scalability problems?

9) Judging by Fig. 3, B, it seems like the algorithm consistently underestimates mutation rates for larger numbers of mutations. Can it be explained? If such biases are possible, how they will affect the statistical comparison of CBN models?

10) How statistically significant is the fact that the subtype-specific H-CBN2 posets share some relations?

**Have all data underlying the figures and results presented in the manuscript been provided?**

Reviewer #1: Yes

Reviewer #2: Yes

PLOS authors have the option to publish the peer review history of their article (what does this mean?). If published, this will include your full peer review and any attached files.

Reviewer #1: No

Reviewer #2: No
---

## [Decision Letter · Decision Letter 1]

9 Aug 2021

Dear Prof. Beerenwinkel,

We are pleased to inform you that your manuscript 'Comparing mutational pathways to lopinavir resistance in HIV-1 subtypes B versus C' has been provisionally accepted for publication in PLOS Computational Biology.

Please make the stylistic changes requested by Rev 2 before submitting your manuscript to production.

Best regards,

Sergei L. Kosakovsky Pond, PhD

Associate Editor

PLOS Computational Biology

Thomas Leitner

Deputy Editor

PLOS Computational Biology

Reviewer's Responses to Questions

**Comments to the Authors:**

Reviewer #2: My comments were addressed satisfactorily. I have just a couple of minor issues:

1) Bars in Fig. 2A,B for larger numbers of mutations are unreadable

2) For the sake of completeness, it makes sense to include the results of comparison of H-CBN2 and MC-CBN for \\varepsilon = 0.1

**Have the authors made all data and (if applicable) computational code underlying the findings in their manuscript fully available?**

Reviewer #2: Yes

PLOS authors have the option to publish the peer review history of their article (what does this mean?). If published, this will include your full peer review and any attached files.

Reviewer #2: No

---

## [Editor Report · Acceptance letter]

2 Sep 2021

PCOMPBIOL-D-20-01701R1 

Comparing mutational pathways to lopinavir resistance in HIV-1 subtypes B versus C

Dear Dr Beerenwinkel,

I am pleased to inform you that your manuscript has been formally accepted for publication in PLOS Computational Biology. Your manuscript is now with our production department and you will be notified of the publication date in due course.

With kind regards,

Livia Horvath
